# CoDA: Coordinated Diffusion Noise Optimization for Whole-Body Manipulation of Articulated Objects

Huaijin Pi[1]    Zhi Cen[2]    Zhiyang Dou[1]    Taku Komura[1]

[1]The University of Hong Kong  [2]Zhejiang University
https://phj128.github.io/page/CoDA

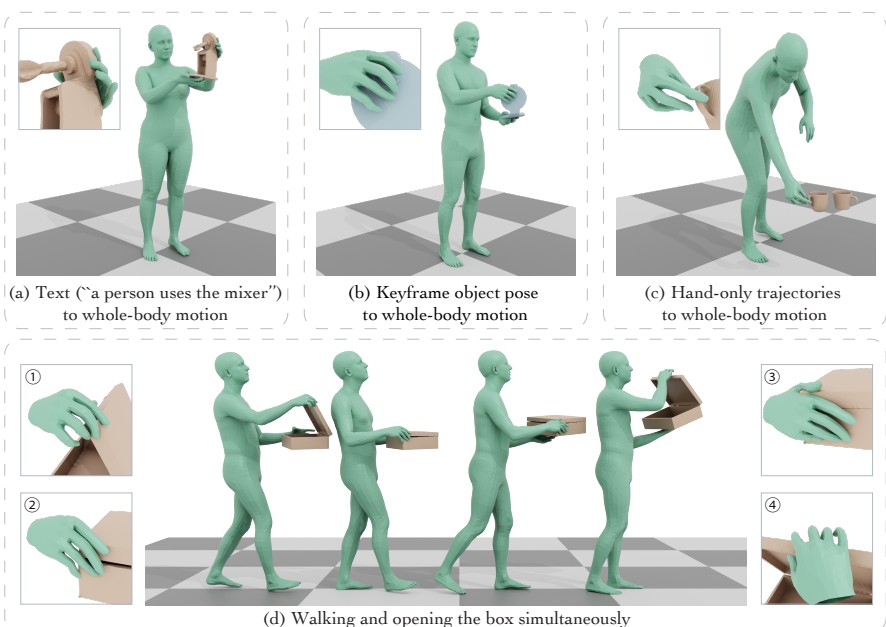

(a) Text (``a person uses the mixer'') to whole-body motion

(b) Keyframe object pose to whole-body motion

(c) Hand-only trajectories to whole-body motion

(d) Walking and opening the box simultaneously

Figure 1: Our approach enables: (a) generating whole-body manipulation of articulated objects from text input (e.g., "a person uses the mixer"); (b) manipulating the object to a target pose and articulation (the blue object is the target pose); (c) synthesizing whole-body motion guided by trajectories from hand-only data; (d) generating motions involving simultaneous walking and object manipulation (e.g., opening a box while walking).

## Abstract

Synthesizing whole-body manipulation of articulated objects, including body motion, hand motion, and object motion, is a critical yet challenging task with broad applications in virtual humans and robotics. The core challenges are twofold. First, achieving realistic whole-body motion requires tight coordination between the hands and the rest of the body, as their movements are interdependent during manipulation. Second, articulated object manipulation typically involves high degrees of freedom and demands higher precision, often requiring the fingers to be placed at specific regions to actuate movable parts. To address these challenges, we propose a novel coordinated diffusion noise optimization framework. Specifically, we perform noise-space optimization over three specialized diffusion models for the body, left hand, and right hand, each trained on its own motion dataset to improve generalization. Coordination naturally emerges through gradient flow

39th Conference on Neural Information Processing Systems (NeurIPS 2025).

along the human kinematic chain, allowing the global body posture to adapt in response to hand motion objectives with high fidelity. To further enhance precision in hand-object interaction, we adopt a unified representation based on basis point sets (BPS), where end-effector positions are encoded as distances to the same BPS used for object geometry. This unified representation captures fine-grained spatial relationships between the hand and articulated object parts, and the resulting trajectories serve as targets to guide the optimization of diffusion noise, producing highly accurate interaction motion. We conduct extensive experiments demonstrating that our method outperforms existing approaches in motion quality and physical plausibility, and enables various capabilities such as object pose control, simultaneous walking and manipulation, and whole-body generation from hand-only data. The code will be released for reproducibility.

# 1   Introduction

Human-object interaction (HOI) motion generation [47, 54] has broad applications in virtual reality, character animation [5, 68, 48], and robotics. These interactions range from simple activities like sitting on a chair [54, 47] to more complex tasks involving articulated object manipulation [6, 21], such as opening a box or a microwave. This paper focuses on the challenging setting of whole-body manipulation of articulated objects. Given an initial pose of the human and the object, along with a textual instruction, our goal is to synthesize realistic, physically plausible interaction sequences that involve coordinated body, hand, and articulated object motion.

Most prior works on HOI generation [16, 47, 74, 4, 28, 29, 8] suffer from two key limitations. First, they typically focus on either body-only motion [47, 28, 29] or hand-only manipulation [73, 81, 6, 21, 8]. Although hand-only methods can produce plausible contact behaviors in short-range scenarios, they fail to capture important whole-body dynamics such as bending down, reaching forward, or walking while manipulating objects. Such whole-body behaviors are essential for generating realistic human-object interactions, especially when manipulation is not restricted to a fixed space. Second, most existing works target rigid objects [70, 29, 8], while articulated objects introduce more complex motion patterns and require continuous in-hand adjustments.

Whole-body manipulation of articulated objects is highly challenging. First, it demands coordinated motion between the body and hands to reflect natural physical behaviors. Body movement affects how the hands approach and manipulate objects, and conversely, hand-object interactions can influence global posture. Second, precise control of finger positions is essential to maintain accurate, physically plausible contact throughout the sequence. This is especially important for articulated objects, where the manipulation often requires placing the fingers at specific regions to actuate the articulation while avoid colliding with other parts.

To address these challenges, we propose a novel framework called CoDA (**Co**ordinated **D**iffusion noise optimization for whole-body manipulation of **A**rticulated objects), which jointly synthesizes the motions of the human body, hands, and articulated objects. Our core idea is to optimize the input noise vectors of three specialized diffusion models, which independently model the body, left hand, and right hand, to generate coordinated whole-body motion. This decoupled design allows each component to be trained on its own data source, such as using large-scale datasets like AMASS [38] for body motion, manipulation datasets like ARCTIC [13] and GRAB [56] for hand motion, thereby improving generalization across diverse motions. Coordination naturally emerges during optimization, as gradients from hand motion objectives flow through the human kinematic chain, allowing the global posture to adapt in response to fine-grained hand motion. This optimization further enables precise control over hand-object contact, while the diffusion noise space [25] provides strong motion priors to preserve naturalness in the generated sequences.

To enable precise manipulation while accounting for object geometry and articulation, we adopt a basis point set (BPS) representation [49, 81] to encode both the object surface and end-effector trajectories in a unified form. Specifically, we represent the positions of the end-effectors, namely the wrists and fingertips, by their distances to the same BPS used for encoding the object geometry. The unified representation captures the relative spatial relationship between the hand and the object geometry as well as its articulation during complex manipulation tasks. The generated trajectories, based on this representation, provide a continuous target signal for optimizing whole-body motion.

We evaluate our approach on both the ARCTIC [13] dataset of articulated object manipulation and the GRAB [56] dataset of rigid object interactions. Our method achieves state-of-the-art performance on both benchmarks, outperforming existing approaches in motion quality and physical plausibility. Beyond benchmark evaluation, our framework enables several compelling capabilities, as illustrated in Figure 1. It supports object pose control at specific times, and coordinated whole-body behaviors involving simultaneous locomotion and manipulation, which are absent from the ARCTIC dataset. In addition, our framework allows us to leverage hand-only datasets [2] to generate whole-body motion, enabling broader data usage and generalization. To the best of our knowledge, this is the first work to jointly generate body, hand, and articulated object motions for whole-body manipulation tasks.

## 2 Related work

**Human-object interaction.** Human-object interaction (HOI) generation [54, 28, 9] has received increasing attention due to its potential to enable virtual humans to perform various actions in 3D environments. Early works focus on generating static interactions such as sitting or lying on furniture [54, 16, 82, 84, 80], using either auto-regressive pipelines or whole-sequence generation [61, 62, 40, 1, 86]. Recent methods explore diffusion-based models [22, 47, 27, 4, 74, 23] and apply guidance techniques [11, 19] to improve human-scene contact quality. Beyond static objects, several works consider dynamic objects [70, 71] or generate human motion conditioned on given object trajectories [28, 10]. For example, OMOMO [28] proposes a two-stage framework that first generates wrist trajectories and then completes body motion accordingly. Other approaches [45, 70, 29, 12, 53, 72] jointly generate body and object motion, and incorporate contact-aware guidance into the diffusion process to improve the quality. Another line of research [17, 42, 69, 59, 43, 63] enables physically simulated characters to perform scene-level interactions by learning control policies through environment interaction. These methods mainly focus on navigation and interactions with large-scale objects such as furniture or obstacles. While generating plausible body motion, they ignore finger motion, which is crucial for fine-grained manipulation.

**Hand-object interaction.** ManipNet [77] synthesizes object manipulation given wrist and object trajectories, using multiple representations to model the hand-object relationship. GRIP [58] design a temporal hand-object spatial feature for stable grasping. Some works [87, 33] address the task of denoising noisy hand motion to recover clean interaction sequences. While these methods explore various representations for modeling hand-object spatial relationships, they rely on access to predefined wrist and object trajectories. [85, 81, 79] explore settings where only the object trajectory is provided. CAMS [85] introduces a canonicalized representation to enable precise contact generation. [81, 79] generate manipulation by predicting contact maps as intermediate representations. Other works generate hand and object motion jointly, without relying on predefined trajectories. DiffH2O [8] applies grasp guidance to diffusion models for more coherent hand-object interactions.. Text2HOI [6] employs cascaded diffusion to iteratively refine the results. HOIGPT [21] leverages separate codebooks for hands and objects, and jointly predicts motion and text. Physics-based approaches [7, 78] generate grasping motions through reinforcement learning in simulated environments. Despite their differences, all these methods ignore the body context, resulting in floating hand motions.

**Whole-body interaction.** Although there are several whole-body manipulation datasets [56, 13, 24, 23, 76, 37, 34], only a few works consider body and hand interaction simultaneously. [57, 66] assume the object is static and only synthesize approaching and grasp motion. IMoS [14] demonstrates full-body manipulation with given finger motion; it generates body motion auto-regressively and optimizes object trajectories by assuming a static hand-object contact frame. TOHO [30] synthesizes whole-body interactions using implicit representations [18], relying on the same contact assumption to recover object motion. DiffGrasp [83] generates whole-body motion conditioned on given object trajectories using diffusion models, and introduces hand-object guidance to improve interaction quality. Wu et.al. [67] employs LLM [41] to analyze the scene and plan motions for grasping and relocating rigid objects. Other works [64, 65, 75] employ physics-based tracking to mimic manipulation behaviors. [3, 36, 31] explores humanoid grasping, but the generated motions remain unnatural and do not involve complex manipulation. Most of the above methods focus exclusively on rigid object interaction and do not address articulated objects. Compared to rigid object interaction, articulated object manipulation is more complex, as it often requires placing the fingers at specific regions to actuate the articulation.

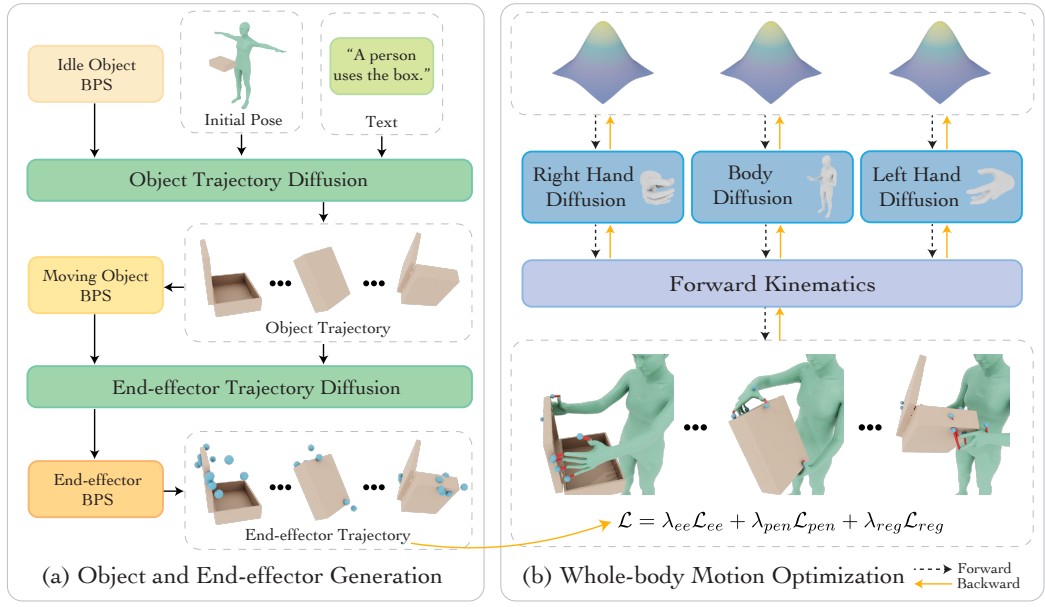

Figure 2: **Pipeline overview.** (a) Given the initial human pose, object pose, and text, we first generate the articulated object trajectory and the corresponding end-effector trajectories via two conditional diffusion models. (b) We then optimize the latent noise inputs of three decoupled diffusion models by propagating gradients through the kinematic chain, guided by end-effector tracking, penetration, and regularization losses. Finally, we forward the optimized noise through the diffusion models to synthesize coherent whole-body motion aligned with the generated object motion.

## 3 Preliminary

In this section, we define the input and output in this paper. Given the initial pose of a human and an articulated object, along with a textual instruction, our goal is to generate a full sequence including the whole-body human motion (body and fingers) and the articulated object motion over time.

**Object representations.** The objects from the ARCTIC [13] dataset are two-part articulated objects with 7 degrees of freedom. We use $\mathbf{S}_o = \{\mathbf{T}_o, \mathbf{R}_o, \mathbf{a}\}$ to indicate the object pose, where the object state $\mathbf{S}_o \in \mathbb{R}^7$ consists of object translation $\mathbf{T}_o \in \mathbb{R}^3$, object rotation $\mathbf{R}_o \in \mathbb{R}^3$, and the angle of the rotational joint $\mathbf{a} \in \mathbb{R}^1$ between the two parts of the object.

**Motion representations.** We use SMPL-X [44], which is a parametric human body model to represent the whole body, including the face and fingers. SMPL-X is a differentiable function that takes input shape, pose, and expression parameters and outputs a 3D mesh with $10,475$ vertices and $20,908$ triangles. The vertices are posed with linear blend skinning with a rigged skeleton which is learned from the data. As we focus on the body motion with two hands, we remove the face related parameters. $\mathbf{\Theta} = \{\boldsymbol{\theta}, \mathbf{t}\}$ is the pose parameters to drive the SMPL-X model, where $\boldsymbol{\theta} \in \mathbb{R}^{52 \times 3}$ represents joint angles and $\mathbf{t} \in \mathbb{R}^3$ is the root translation.

**Text descriptions.** In the ARCTIC [13] and the GRAB [56] dataset, each sequence is annotated with an action label. Following previous work [14, 6], we construct the text description using the template "A person <action> the <object>.". For example, "A person uses the box.".

## 4 Method

The overview of our pipeline is shown in Figure 2. We first generate the motion of the articulated object (Section 4.1), then predict the end-effector trajectories (Section 4.2), and finally synthesize the whole-body motion by optimizing the noise of decoupled diffusion models (Section 4.3).

## 4.1 Object motion generation

Given the initial object pose and the textual instruction, we train a diffusion model [60] to generate the object future trajectory. The input includes the CLIP [50] feature of the text, the initial object pose, and the object geometry embedding. We represent the object geometry using the normalized part-based BPS descriptor [81], which will be formally defined in Section 4.2 and Figure 3. The output is a sequence of object states over time.

## 4.2 End-effector trajectory generation

Given the generated object trajectory, we extract its geometry representation and combine it with the trajectory itself and the textual instruction as input to a diffusion model that predicts end-effector trajectories. Instead of directly predicting 3D joint coordinates [28], we design a distance-based representation that encodes end-effector positions in the same space as the object geometry.

**Unified BPS-based representation for object and end-effectors.** We first present the object geometry representation. Following previous work [81], we adopt the normalized part BPS [49] to represent the object geometry. Specifically, the object mesh is first normalized to the unit scale by dividing all vertex coordinates by the maximum distance from the object origin to any vertex. Then a pre-defined fixed set of basis points $\mathbf{P} \in \mathbb{R}^{K \times 3}$, shared across all objects, are uniformly sampled within the unit sphere centered at the object origin. The BPS representation is computed as the distances from each basis point to the nearest vertex on each of the two rigid object parts, resulting in an object geometry vector $\mathbf{O} \in \mathbb{R}^{K \times 2}$.

We then introduce *end-effector BPS*, a distance-based representation tailored for encoding the positions of end-effectors in the object coordinate system. The end-effectors include both wrists and fingertips, comprising a total of 12 joints (2 wrists and 10 fingertips). As shown in figure 3, at each frame, for each of the 12 end-effectors, we compute a $K$-dimensional vector of Euclidean distances to the basis points. We use the same pre-defined set of basis points $\mathbf{P} \in \mathbb{R}^{K \times 3}$ in object geometry representation [81]. This results in a $(12 \times K)$-dimensional end-effector BPS vector per frame. The diffusion model outputs a sequence of end-effector BPS over time, along with binary contact labels for each fingertip, indicating whether it is close to the object surface.

Given the generated end-effector BPS sequence, we recover the end-effector trajectories by solving a simple optimization problem. For each end-effector at each frame, we minimize the following loss to infer its 3D position:

$$\mathbf{p}_e^* = \arg\min_{\mathbf{p}_e} \mathcal{L}(\mathbf{p}_e), \quad (1)$$

$$\mathcal{L}(\mathbf{p}_e) = \sum_j \|\|\mathbf{p}_e - \mathbf{P}_j\|_2 - d_j\|_2, \quad (2)$$

where $\mathbf{p}_e$ is the optimized 3D position and $d_j$ is the predicted distance to the $j$-th basis point $\mathbf{P}_j$. By sharing the basis point set with the object BPS representation, our method provides a consistent spatial reference frame that facilitates geometric alignment between end-effectors and object parts.

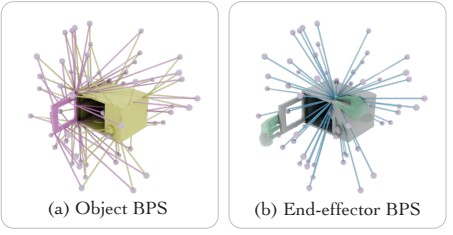

(a) Object BPS      (b) End-effector BPS

Figure 3: **The illustration of the end-effector BPS.** (a) is the object BPS [81]. (b) is the proposed end-effector BPS representation. Gray points denote the basis points; pink/yellow are two object parts; blue indicates a fingertip. Only one end-effector and 64 basis points are visualized for simplicity.

**RoPE-based object motion encoding.** To better encode the object trajectory, we adapt the idea of CaPE [26], which encodes relative camera pose information via RoPE [55]. In our case, each object pose is also represented as a $4 \times 4$ transformation matrix. Inspired by CaPE [26], we use the object pose to transform the query and key features in each attention layer. This enables the model to encode the relative object motion within a local temporal window, providing temporally-aware conditioning for the generation. We refer readers to the supplementary material for more details.

## 4.3 Whole-body motion generation

The goal of this stage is to generate coherent whole-body motion that aligns with the predicted end-effector trajectories and articulated object motion. Rather than directly predicting whole-body

poses conditioned on end-effectors [28], we adopt an optimization-based approach inspired by DNO [25]. Specifically, we optimize the noise input to the diffusion models (Figure 2 (b)), and then forward the optimized noise through the diffusion models to generate the final motion. To further improve motion quality, we decouple the body into three components: body, left hand, and right hand, and train separate diffusion models for each. This decoupled design enables us to train each module using individual data, such as training the hand models using the ARCTIC [13] and GRAB [56], and the body-only model without hands on the AMASS [38]. Such specialization improves generalization by allowing novel combinations of finger and body motion to be synthesized. Moreover, this formulation facilitates gradient flow through the kinematic chain during optimization, which improves coordination between the body and hands.

**Decoupled motion diffusion model.** We adopt a decoupled human representation for whole-body motion, dividing the human pose into three components: body, left hand, and right hand. Formally, for each frame $i$, the whole-body pose $\boldsymbol{\Theta}_i$ is represented as:

$$\mathbf{x} = \{\mathbf{x}_b, \mathbf{x}_{lh}, \mathbf{x}_{rh}\}, \tag{3}$$

$$\mathbf{x}_b = \{\dot{r}^x, \dot{r}^z, r^y, \dot{r}^a, \boldsymbol{\theta}_b\}, \tag{4}$$

$$\mathbf{x}_{lh} = \{\boldsymbol{\theta}_{lh}\}, \tag{5}$$

$$\mathbf{x}_{rh} = \{\boldsymbol{\theta}_{rh}\}, \tag{6}$$

where $x_b$ denotes the body component, including root velocities $\dot{r}^x, \dot{r}^z \in \mathbb{R}$ (projected on the XZ-plane), root height $r^y \in \mathbb{R}$, angular velocity $\dot{r}^a \in \mathbb{R}$, and body joint rotations $\boldsymbol{\theta}_b \in \mathbb{R}^{6 \times J_b}$, while $\boldsymbol{\theta}_{lh} \in \mathbb{R}^{6 \times J_{lh}}$ and $\boldsymbol{\theta}_{rh} \in \mathbb{R}^{6 \times J_{rh}}$ represent the left and right hand joint rotations, respectively. All joint rotations are encoded using the 6D representation [88], with $J_b = 22$, $J_{lh} = J_{rh} = 15$ joints for the body and each hand. We train three separate diffusion models, $\mathbf{M}_b$, $\mathbf{M}_{lh}$, and $\mathbf{M}_{rh}$, to model the motion manifolds of the body and hands individually.

**Optimization over diffusion noise.** Given the trained diffusion models for body, left hand, and right hand, we optimize the noise vectors $\mathbf{z} = \{\mathbf{z}_b, \mathbf{z}_{lh}, \mathbf{z}_{rh}\}$ to generate whole-body motion as shown in Figure 2 (b). Let $f(\mathbf{z})$ denote the process that maps the input noise to global joint positions through diffusion models and forward kinematics:

$$f(\mathbf{z}) = \mathcal{FK}(\mathbf{M}_b(\mathbf{z}_b), \mathbf{M}_{lh}(\mathbf{z}_{lh}), \mathbf{M}_{rh}(\mathbf{z}_{rh})), \tag{7}$$

where $\mathcal{FK}(\cdot)$ converts root translation and local joint rotations into global joint positions. We formulate motion generation as minimizing a loss $\mathcal{L}$ over the diffusion noise:

$$\mathbf{z}^* = \arg\min_{\mathbf{z}} \mathcal{L}(f(\mathbf{z})). \tag{8}$$

The overall loss function consists of three components with different weights $\lambda_{ee}$, $\lambda_{pen}$, and $\lambda_{reg}$:

$$\mathcal{L} = \lambda_{ee}\mathcal{L}_{ee} + \lambda_{pen}\mathcal{L}_{pen} + \lambda_{reg}\mathcal{L}_{reg}, \tag{9}$$

where $\mathcal{L}_{ee}$, $\mathcal{L}_{pen}$, and $\mathcal{L}_{reg}$ are the end-effector tracking, penetration, and regularization losses.

We encourage the generated global fingertip positions $\hat{\mathbf{p}}_f$ to follow the predicted trajectories $\mathbf{p}_f$ from the previous stage. We also constrain the relative fingertip positions to the wrist joints:

$$\mathcal{L}_{ee} = \|\hat{\mathbf{p}}_f - \mathbf{p}_f\|_1 + \|\hat{\mathbf{p}}_f^r - \mathbf{p}_f^r\|_1, \tag{10}$$

where $\mathbf{p}_f^r$ and $\hat{\mathbf{p}}_f^r$ denote the relative fingertip positions with respect to the wrist.

To reduce hand-object interpenetration, we penalize fingertip joints that fall inside the object mesh:

$$\mathcal{L}_{pen} = \sum_j \left\| \min\left(\mathrm{SDF}(\mathbf{J}^j) - 0.01, 0.00\right) \right\|_1 \tag{11}$$

where $\mathrm{SDF}(\mathbf{J}^j)$ is the signed distance at the $j$-th hand joint, assuming 1cm finger thickness.

We add a regularization term to discourage foot floating and foot sliding:

$$\mathcal{L}_{reg} = \left\| \min\left(\mathbf{J}^y\right) - 0.02 \right\|_1 + \mathbf{1}_{\text{left}} \cdot \left\| \mathbf{J}_i^l - \mathbf{J}_{i-1}^l \right\|_1 + \mathbf{1}_{\text{right}} \cdot \left\| \mathbf{J}_i^r - \mathbf{J}_{i-1}^r \right\|_1, \tag{12}$$

where $\mathbf{J}^y$ denotes the height of all joints in the body, and $\mathbf{J}_i^l$ and $\mathbf{J}_i^r$ denote the 3D positions of the left and right foot joints at frame $i$, respectively. The binary indicators $\mathbf{1}$ denote whether the left or right foot is in contact with the ground, based on a height threshold of $0.02$ meters.

We adopt DDIM [52] sampling to efficiently generate motion sequences during optimization following DNO [25]. The loss is computed on the final output, and gradients are propagated back through the DDIM solver to update the noise. After optimization, we pass the optimized noise into the decoupled diffusion models to generate the final whole-body motion. Combined with the previously generated object trajectory, this yields a complete human-object manipulation sequence. This noise-space optimization avoids high-dimensional pose regression, reduces artifacts, and produces natural whole-body motions aligned with the object manipulation process.

## 5 Experiments

### 5.1 Implementation details

We adopt a transformer-based diffusion architecture similar to MDM [60] for all models in our framework. During inference, we perform noise optimization using DDIM [52] with $T=10$ for 800 steps and a cosine-decayed learning rate, following the DNO [25] strategy. All experiments are conducted on a single NVIDIA A100 GPU. More training details are in the supplementary material.

### 5.2 Dataset and evaluation metrics

**Dataset.** We evaluate on ARCTIC [13] for articulated object manipulation and on GRAB [56] for rigid object interaction. ARCTIC contains around 2 hours of motion data featuring 10 subjects interacting with 11 articulated objects, including complex motions such as bimanual grasps and in-hand manipulation. Following the protocol in [81], we randomly sample 4 sequences per object category to construct the test set. The GRAB dataset covers about 4 hours of interaction from 10 subjects with 51 rigid objects, focusing primarily on grasping and simple lifting actions. Similar to [14], we use data from the last subject as the test set. For training object motion and end-effector trajectories generation, ARCTIC is used for articulated objects, and GRAB is used for rigid objects. The body motion model is trained on ARCTIC, GRAB, and AMASS [38], while the two hand motion models are trained on ARCTIC and GRAB.

**Evaluation metrics.** Similar to [8, 6], we evaluate the motion quality using the following metrics: (1) **Frechet Inception Distance (FID)** measures the feature-level distance between generated and real motions, using a motion feature extractor trained on the dataset following [15]. (2) **R-Precision** quantifies the alignment between generated motion and the corresponding textual prompt, measured using Top-3 accuracy. (3) **Diversity** reflects the variation among generated motion samples. (4) **Foot skating** indicates motion realism by detecting undesired foot sliding, following the computation in [32, 47]. We additionally report physical realism metrics following [8]: (5) **Interpenetration volume (IV)** computes the number of hand vertices that penetrate the object mesh. (6) **Interpenetration depth (ID)** measures the maximum penetration depth of hand vertices into the object. (7) **Contact ratio (CR)** is defined as the average proportion of hand vertices within 5 mm of the object surface. We also conduct a user study involving 16 participants to evaluate the generated motion sequences.

### 5.3 Comparison with baselines

**Baselines.** As there is no existing method that jointly generates body, hand, and articulated object motion, we adapt several representative methods to our task: IMoS [14], MDM [60], OMOMO [28], Text2HOI [6], and CHOIS [29]. IMoS is a CVAE-based [51] auto-regressive model, while MDM is a full-sequence diffusion-based [20] model. Text2HOI is originally designed for hand-object interaction with multiple diffusion models for iterative refinement. CHOIS is a diffusion-based model that incorporates contact guidance during inference. We extend them to jointly generate whole-body motion and object motion. OMOMO first generates wrist motion and then synthesizes body motion. We extend it to a three-stage model: first generating object motion, then predicting fingertip and wrist trajectories, and finally producing whole-body motion. OMOMO+DNO further extends OMOMO by using its diffusion model as a latent prior and applying DNO [25] to refine the generated results.

**Quantitative results.** We report quantitative results on ARCTIC and GRAB in Table 1 and Table 3, and user study results in Table 2. Our method achieves the best performance on nearly all metrics across both datasets. While it ranks slightly lower in diversity, it significantly outperforms all

Table 1: **Comparison on the ARCTIC [13] dataset.** The right arrow → means the closer to real motion the better. IV, ID, and CR denote interpenetration volume, interpenetration depth, and contact ratio. The best and second-best results are highlighted green and yellow, respectively.

| Methods | FID↓ | R-Precision↑ | Diversity→ | Foot skating↓ | IV↓ | ID↓ | CR↑ |
|---|---|---|---|---|---|---|---|
| Real | – | 0.531 | 8.664 | 0.002 | 4.68 | 11.47 | 0.085 |
| IMoS [14] | 6.686 | 0.305 | 6.144 | 1.469 | 14.28 | 13.24 | 0.010 |
| MDM [60] | 3.972 | 0.209 | 8.167 | 0.027 | 16.90 | 15.85 | 0.033 |
| Text2HOI [6] | 6.654 | 0.234 | 5.923 | 0.028 | 12.72 | 17.14 | 0.010 |
| OMOMO [28] | 3.710 | 0.406 | 6.110 | 0.028 | 13.77 | 15.16 | 0.061 |
| OMOMO + DNO [25] | 2.873 | 0.391 | 7.004 | 0.022 | 8.95 | 13.30 | 0.075 |
| CHOIS [29] | 3.758 | 0.367 | 7.423 | 0.023 | 17.19 | 15.84 | 0.030 |
| Ours | 2.283 | 0.477 | 7.208 | 0.002 | 5.25 | 12.87 | 0.086 |

Table 2: **User study on the ARCTIC [13] dataset.**

| Metrics | Ours | CHOIS [29] | OMOMO [28] | Text2HOI [6] |
|---|---|---|---|---|
| Best Motion Realism Rate ↑ | 88.7% | 1.1% | 9.9% | 0.3% |
| Best Physical Plausibility Rate ↑ | 87.3% | 1.4% | 10.2% | 1.1% |

baselines in the user study, indicating superior perceptual quality and physical plausibility. To assess the physical feasibility of our method, we conduct a mimic-based evaluation following [46, 64], where a humanoid policy is trained to reproduce the generated motions in the IsaacGym [39] simulator. We use eight generated sequences (each 10 seconds long) involving boxes and microwaves, and measure the tracking duration—the time (in seconds) during which both object and joint position errors remain below a 10 cm threshold. Results in Table 4 show that our motions lead to longer tracking durations compared to OMOMO, demonstrating improved physical plausibility and better compatibility with downstream humanoid execution. In addition, to evaluate the generalization capability of our method, we conduct an additional experiment on the ARCTIC dataset by holding out the box object for testing and training on the remaining objects. As shown in Table 5, our method achieves a substantially lower FID compared to the baseline, demonstrating better object-level generalization.

**Qualitative results.** As demonstrated in Figure 4, our method achieves significantly better hand-object contact compared to baselines. We provide more results in the supplementary material.

## 5.4 Ablation study

We ablate key components of our framework to understand their impact on overall performance: (a) A single model to jointly predict object motion and end-effector trajectories. (b) Predicting relative coordinate of end-effectors to the object center without end-effector BPS. (c) Using object velocity and rotational velocity as the trajectory input without RoPE-based representation. (d) Removing the optimization process and using a conditional diffusion model with fingertip trajectories as input. (e) Using a single diffusion model for the entire body without the decoupled body-hand representation. (f) Excluding the AMASS [38] dataset during training the body motion model. (g) replacing the end-effector representation with a distance field [77], where the trajectory is encoded as a fixed grid of distances in the object's local coordinate frame, while keeping the object geometry encoded using

Table 3: **Comparison on the GRAB [56] dataset.**

| Methods | FID↓ | R-Precision↑ | Diversity→ | Foot skating↓ | IV↓ | ID↓ | CR↑ |
|---|---|---|---|---|---|---|---|
| Real | – | 0.727 | 15.045 | 0.010 | 5.84 | 13.41 | 0.049 |
| IMoS [14] | 52.290 | 0.180 | 8.374 | 0.152 | 11.57 | 20.35 | 0.000 |
| MDM [60] | 26.734 | 0.289 | 8.627 | 0.109 | 12.96 | 16.03 | 0.001 |
| Text2HOI [6] | 30.101 | 0.320 | 10.302 | 0.086 | 12.52 | 14.55 | 0.000 |
| OMOMO [28] | 25.017 | 0.391 | 9.294 | 0.094 | 11.03 | 14.03 | 0.004 |
| CHOIS [29] | 25.835 | 0.320 | 9.887 | 0.055 | 9.31 | 14.37 | 0.002 |
| Ours | 21.544 | 0.438 | 9.387 | 0.046 | 4.93 | 10.23 | 0.040 |

Table 4: **Physical feasibility evaluation on the ARCTIC [13] dataset.**

| Metrics | OMOMO [28] | Ours |
|---|---|---|
| Tracking Duration ↑ | 4.75 | 8.75 |

Table 5: **Comparison on the ARCTIC [13] dataset with held-out box object.**

| Methods | FID↓ | R-Precision↑ | Diversity→ | Foot skating↓ | IV↓ | ID↓ | CR↑ |
|---|---|---|---|---|---|---|---|
| OMOMO [28] | 44.009 | 0.547 | 6.234 | 0.028 | 26.92 | 8.23 | 0.116 |
| Ours | 16.091 | 0.547 | 4.964 | 0.002 | 26.24 | 8.56 | 0.128 |

BPS. (h) conditioning the hand motion diffusion model on object trajectories. As shown in Table 6, each component contributes to the performance improvement.

## 5.5 More discussions

**Generalization to different object geometry.** To further validate generalization to unseen object geometries of the same category, we train the object motion and end-effector trajectory models on the hand-only dataset [85], using 7 training and 3 testing objects. Despite the dataset containing only hand motion, our method successfully generates whole-body motion, as shown in Figure 5.

**Various capabilities.** Our approach enables various capabilities. First, it allows control over keyframe object poses by setting them as optimization targets for object trajectory generation. Second, it can synthesize whole-body motions that involve simultaneous locomotion and manipulation, even though such combinations are not present in the training dataset [13]. Third, it enables generating whole-body motion guided by hand-only datasets [2], using wrist and fingertip trajectories as optimization targets. We provide more qualitative results in the supplementary material.

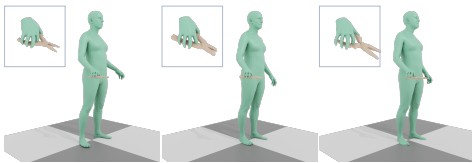

Figure 5: **Generalization to different object geometry.** We train the object motion and end-effector trajectory models on hand-only data [85] with diverse object geometries. These models are integrated into our framework to provide optimization targets, enabling realistic whole-body motion synthesis for unseen object geometry.

**Deployment on simulated humanoids** As shown in Figure 6, our generated whole-body motion can serve as a reference for controlling humanoids in physics-based simulators. We apply physical motion tracking methods [46, 35, 64] to track the synthesized motions. The humanoid is able to physically interact with objects and perform coordinated manipulation behaviors in the simulated environment.

**Inference speed** We report the inference time of each module in our pipeline, measured on a single NVIDIA A100 GPU for generating a 300-frame motion sequence. The object motion model

Table 6: **Ablation study on the ARCTIC [13] dataset.**

| Methods | FID↓ | R-Precision↑ | Diversity→ | Foot skating↓ | IV↓ | ID↓ | CR↑ |
|---|---|---|---|---|---|---|---|
| Real | – | 0.531 | 8.664 | 0.002 | 4.68 | 11.47 | 0.085 |
| (a) w/o separate models | 3.790 | 0.438 | 6.939 | 0.002 | 8.21 | 13.16 | 0.103 |
| (b) w/o end-effector BPS | 4.069 | 0.453 | 6.888 | 0.002 | 8.09 | 13.54 | 0.093 |
| (c) w/o RoPE motion | 2.714 | 0.469 | 7.021 | 0.002 | 6.12 | 12.66 | 0.093 |
| (d) w/o optimization | 4.883 | 0.414 | 6.406 | 0.030 | 16.39 | 16.13 | 0.095 |
| (e) w/o decoupled | 2.699 | 0.438 | 7.142 | 0.008 | 12.45 | 16.29 | 0.082 |
| (f) w/o AMASS | 3.305 | 0.453 | 6.859 | 0.003 | 5.46 | 13.04 | 0.089 |
| (g) w/ SDF | 2.350 | 0.422 | 7.064 | 0.003 | 6.69 | 13.97 | 0.086 |
| (h) w/ extra condition | 2.998 | 0.453 | 7.212 | 0.002 | 8.95 | 13.30 | 0.075 |
| Ours | 2.283 | 0.477 | 7.208 | 0.002 | 5.25 | 12.87 | 0.086 |

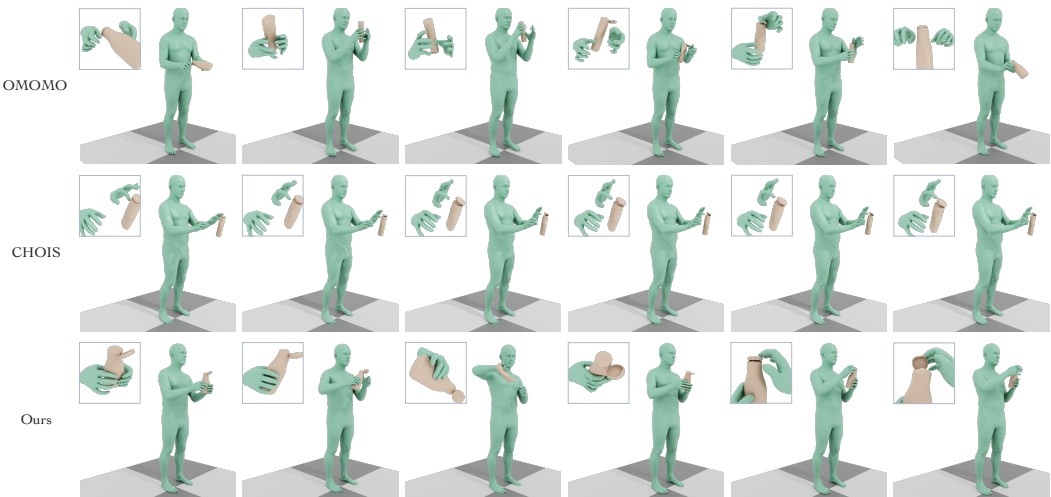

Figure 4: **Qualitative comparison.** Given the text "A person uses the ketchup.", our method generates the whole-body motion with better hand-object contact compared to baselines.

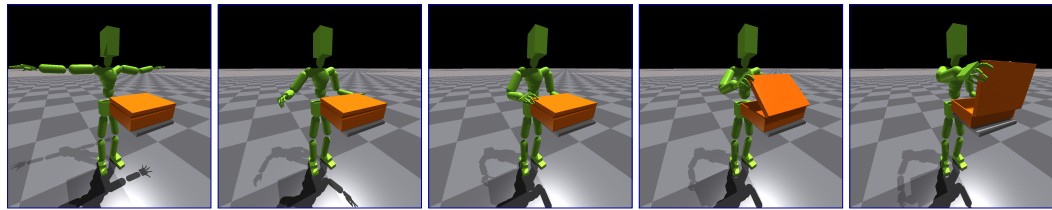

Figure 6: **Deployment on simulated humanoids.** We apply existing motion tracking techniques to deploy the generated motion to a simulated humanoid. The articulated object is physically manipulated by the humanoid within the physics simulator [39].

requires approximately 0.52 seconds, the end-effector model takes about 3.66 seconds, and the whole-body motion optimization, which involves iterative diffusion sampling and gradient-based updates, takes around 16.9 minutes. Most of the computation time is spent on the whole-body optimization stage. Although slower than feed-forward approaches such as CHOIS [29], this optimization process produces motions with substantially higher quality and physical plausibility.

**Limitations.** First, the optimization process is slower than other generative methods [60], limiting real-time applications. Second, due to the limited object diversity in existing datasets [13], the model struggles to generalize to novel object categories. Third, our framework only focuses on single-object manipulation; extending it to handle multiple interacting objects or multi-step sequential interactions remains an open direction. Finally, enabling both the body and fingers to reason about and avoid obstacles in complex scenes, such as surrounding geometry or other objects, is still a difficult problem.

## 6 Conclusion

In this paper, we present a coordinated diffusion noise optimization framework for synthesizing whole-body manipulation of articulated objects. By optimizing over the noise space of separately trained diffusion models for the body, left hand, and right hand, our method enables natural coordination between the body and hands. We introduce a unified distance-based representation built on basis point sets to generate end-effector trajectories, facilitating precise hand-object interactions. Extensive experiments demonstrate that our approach achieves state-of-the-art performance in motion quality and physical plausibility. It also supports various capabilities such as object pose control, simultaneous manipulation and locomotion, and whole-body motion generation from hand-only data.

## 7  Acknowledgments

This work is partly supported by the Innovation and Technology Commission of the HKSAR Government under the ITSP-Platform grant (Ref: ITS/335/23FP) and the InnoHK initiative (TransGP project). Part of the research was conducted in the JC STEM Lab of Robotics for Soft Materials, funded by The Hong Kong Jockey Club Charities Trust.

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
