# OpenReview forum: "CoDA: Coordinated Diffusion Noise Optimization for Whole-Body Manipulation of Articulated Objects"
_NeurIPS.cc/2025/Conference — NeurIPS 2025 poster_

### Official Review · Reviewer_4399 · 2025-06-27

**Clarity:** 4
**Significance:** 2
**Originality:** 2
**Rating:** 4
**Confidence:** 3

**Summary:**

This paper proposes a novel framework for synthesizing whole-body manipulation of articulated objects by coordinating body, hand, and object motions. The approach leverages three separately trained diffusion models for the body, left hand, and right hand, and performs joint noise-space optimization to enable coordination through the human kinematic chain. A unified object-hand representation using basis point sets (BPS) enhances fine-grained control and precision in interaction. The method supports capabilities such as object pose control, walking during manipulation, and motion synthesis from hand-only data. Experiments show improved motion quality and physical plausibility over prior work.

**Questions:**

1. Articulated object manipulation has been extensively studied in the robotics community. Could the authors elaborate on how their approach relates to prior work in robotics, particularly those involving interaction with complex mechanisms (e.g., PartNet-Mobility, AKB-48, DexArt, AdaManip)? A more thorough discussion would help clarify the contribution relative to existing manipulation pipelines.

2. Given the limitations of the ARCTIC dataset and the use of static HOI datasets and perceptual metrics (e.g., FID, R-Precision), the current evaluation may not fully reflect the model’s effectiveness in physically grounded scenarios. Have the authors considered evaluating their method in interactive simulation environments such as SAPIEN or Isaac Sim? Benchmarking the success rate of generated motions in these settings would provide more robust validation. As noted in the limitation section of the paper, extending the framework to support sequential interactions remains an open challenge—one that datasets like AdaManip are designed to capture.

3. Have the authors considered validating their method on a real-world humanoid robot? While simulation is a useful intermediary, demonstrating that the generated motions can transfer to physical platforms would significantly strengthen the practical relevance of this work.

**Ethical Concerns:**

["NO or VERY MINOR ethics concerns only"]

**Final Justification:**

The authors' rebuttal addresses most of my concerns, and I am raising my rating to 4: Borderline Accept. However, a higher rating is not warranted at this stage, as the evaluation of more realistic and complex articulated object manipulation, both in simulation and on real humanoid robots, are left for future work.

**Limitations:**

See the Questions section. The paper would benefit from additional evaluation on more diverse and realistic simulation benchmarks and real-world experiments to better demonstrate the method's generalizability and practical applicability.

**Paper Formatting Concerns:**

No formatting concerns.

**Quality:**

3

**Strengths And Weaknesses:**

**Strengths**
1. The paper tackles the problem of whole-body human-object interaction, which is more realistic than hand-only interaction. The proposed method decomposes the task into object trajectory generation, end-effector trajectory generation, and coordinated diffusion noise optimization via the kinematic chain. This modular design improves joint control and generalization. The use of forward kinematics enables automatic optimization and adaptability across different humanoid configurations.

2. The paper is well-written and clearly presented, with high-quality visualizations that effectively convey the model’s capabilities.

3. The authors provide comprehensive comparisons against relevant baselines and include meaningful ablation studies to validate key design choices.

**Weaknesses**
1. The evaluation is limited to static HOI datasets. The work would be strengthened by demonstrating performance in simulation environments (e.g., SAPIEN, Isaac Sim) or on real-world humanoid robots to assess applicability in dynamic, embodied settings.

2. The articulated objects used (from the ARCTIC dataset) are relatively simple two-part structure, which may not capture the complexity of real-world articulated objects. Evaluation on more diverse datasets such as PartNet-Mobility[1], AKB-48[2], DexArt[3], or AdaManip[4] would improve the generality of the conclusions. This limitation is also acknowledged by the authors.

3. Several components of the proposed framework (e.g., the BPS-based representation, RoPE-style encoding from CaPE, and DDIM sampling from DNO) are adaptations of existing techniques. While the integration is novel, the method may be viewed as an incremental combination rather than a fundamentally new paradigm.

[1] Xiang, Fanbo, et al. "Sapien: A simulated part-based interactive environment." Proceedings of the IEEE/CVF conference on computer vision and pattern recognition. 2020.

[2] Liu, Liu, et al. "Akb-48: A real-world articulated object knowledge base." Proceedings of the IEEE/CVF Conference on Computer Vision and Pattern Recognition. 2022.

[3] Bao, Chen, et al. "Dexart: Benchmarking generalizable dexterous manipulation with articulated objects." Proceedings of the IEEE/CVF Conference on Computer Vision and Pattern Recognition. 2023.

[4] Wang, Yuanfei, et al. "Adamanip: Adaptive articulated object manipulation environments and policy learning." arXiv preprint arXiv:2502.11124 (2025).

---

> ### Author Rebuttal · Authors · 2025-07-29
>
> ## **Response to Reviewer 4399**
> Thank you for your thoughtful feedback and constructive suggestions. We are glad that you find our paper $\textcolor{blue}{\text{well-written and clearly presented}}$, with $\textcolor{blue}{\text{high-quality visualizations}}$, $\textcolor{blue}{\text{comprehensive comparisons}}$, and $\textcolor{blue}{\text{meaningful ablations}}$.
>
> ### **Weakness**
> **Demonstrating performance in simulation environments.**
>
> | Methods      | Tracking Duration↑  |
> |--------------|---------------------|
> | OMOMO        | 4.75 secs           |
> | Ours         | 8.75 secs           |
>
> We appreciate the suggestion to evaluate in interactive simulators. We have included deployment on simulated humanoids visualization in $\textcolor{green}{\text{Section E.6}}$ in the supplementary material. To further assess the physical feasibility of our method, we conduct a mimic-based [5, 6] evaluation, where a humanoid policy is trained to mimic the generated motions in the IsaacGym [10] simulator. We use 8 generated sequences (each 10 seconds long) involving boxes and microwaves, and track how long the humanoid controller can successfully mimic the motion. Following [5, 6], the tracking duration is the time (in seconds) that the object and joint error are lower than a threshold of 10 cm. Compared to OMOMO, our motions lead to longer durations, indicating improved physical plausibility and better compatibility with downstream humanoid execution.
>
> **Limited articulation complexity in ARCTIC.**
>
> Thank you for pointing out these articulated object benchmarks from the robotics community. While datasets such as PartNet-Mobility [1], AKB-48 [2], and DexArt [3] contain more complex object geometries, they lack paired human motion data, which is essential for our learning-based framework. As our method relies on motion capture data for training, we are currently limited by the available human-object interaction datasets. We hope future benchmarks will include richer articulated objects along with corresponding human motion data.
>
> **Novel integration without fundamentally new paradigm.**
>
> Our method presents a novel framework tailored for the new and challenging task of generating whole-body manipulation motions for articulated objects. The way we formulate and tackle this problem is itself novel. To meet the unique demands of this setting, we design a tailored framework that integrates existing components in new and effective ways:
> 1. We introduce BPS not just for static object geometry encoding, as in prior work [7,8], but to represent time-varying end-effector trajectories. This allows for fine-grained geometric alignment over the entire motion sequence.
> 2. Our joint noise optimization is not a direct reuse of existing strategies [9]. We coordinate separate diffusion models for the body and both hands by linking their gradients through the human kinematic chain. This enables whole-body motion composition with precise interaction control, which is not explored in previous diffusion-based methods [7, 8, 9].
>
> This perspective is shared by $\textcolor{blue}{\text{Reviewer 3mBR}}$, who described our framework as $\textcolor{blue}{\text{interesting}}$ and our unified representation as $\textcolor{blue}{\text{novel and effective}}$.
>
> ### **Questions**
> **Relation to articulated object manipulation in robotics community.**
>
> Our task setting is fundamentally different from manipulation pipelines in robotics benchmarks such as PartNet-Mobility [1], AKB-48 [2], DexArt [3], and AdaManip [4]. Specifically:
> 1. **Goal**: These works [1, 2, 3, 4] primarily focus on physical execution, ensuring that a robotic agent can complete manipulation tasks in the real world. In contrast, our focus is on content generation [7, 8]: producing diverse, high-quality, and physically plausible whole-body motions that reflect natural human behavior. These goals are complementary: while robotics emphasizes control and execution, our generated motion can be used as kinematic trajectories or reference motions to guide humanoid control and serve as data for downstream planning and policy learning.
> 2. **Task setting**: These works focus on single-arm grippers [1,2,4] or robotic hands [3], often on fixed bases or wheeled robots. In contrast, our method operates on full human bodies with 52 articulated joints, each with 3 degrees of freedom, and a 6-DoF root, resulting in a 159-dimensional motion space. This enables not only bimanual dexterous manipulation but also coordinated whole-body behaviors such as locomotion during interaction.
> 3. **Data source**: These approaches rely on reinforcement learning [1,2,3] or rule-based expert policies [4] in simulation to collect training data. However, due to the high-dimensional action space and complex coordination in whole-body manipulation, it is difficult to design effective rewards or rules for our task. Instead, we leverage motion capture data from real humans to learn realistic behaviors.
>
> **Evaluation in simulation environments and sequential interactions.**
>
> As noted above, we have conducted simulation-based evaluations using IsaacGym. Regarding sequential interactions, we agree this is a valuable direction and have also discussed it in $\textcolor{green}{\text{L304}}$ of the main paper. While AdaManip [4] focuses on single-arm grippers and uses rule-based policies that are not directly applicable to natural human motion, its formulation of sequential tasks offers useful inspiration. Although it is challenging to generate realistic whole-body human motions for such tasks using handcrafted rules, we hope future benchmarks will provide sequential human-object interaction motion capture data, which would greatly facilitate progress in this direction.
>
> **Extention to real-world humanoid robots.**
>
> We appreciate the suggestion. While our current focus is motion generation, the synthesized whole-body motions can be used as reference motions for real-world humanoid robots. This opens opportunities for integration with existing humanoid control systems, and we see this as an exciting direction for future work.
>
> ---
> [1] Xiang, Fanbo, et al. "Sapien: A simulated part-based interactive environment." Proceedings of the IEEE/CVF conference on computer vision and pattern recognition. 2020.
>
> [2] Liu, Liu, et al. "Akb-48: A real-world articulated object knowledge base." Proceedings of the IEEE/CVF Conference on Computer Vision and Pattern Recognition. 2022.
>
> [3] Bao, Chen, et al. "Dexart: Benchmarking generalizable dexterous manipulation with articulated objects." Proceedings of the IEEE/CVF Conference on Computer Vision and Pattern Recognition. 2023.
>
> [4] Wang, Yuanfei, et al. "Adamanip: Adaptive articulated object manipulation environments and policy learning." arXiv preprint arXiv:2502.11124 (2025).
>
> [5] Luo, Zhengyi, et al. "Perpetual humanoid control for real-time simulated avatars." Proceedings of the IEEE/CVF International Conference on Computer Vision. 2023.
>
> [6] Wang, Yinhuai, et al. "PhysHOI: Physics-Based Imitation of Dynamic Human-Object Interaction." arXiv preprint arXiv:2312.04393 (2023).
>
> [7] Li, Jiaman, et al. "Object motion guided human motion synthesis." ACM Transactions on Graphics (ToG), 2023.
>
> [8] Li, Jiaman, et al. "Controllable human-object interaction synthesis."  European Conference on Computer Vision, 2024.
>
> [9] Karunratanakul, Korrawe, et al. "Optimizing diffusion noise can serve as universal motion priors." Proceedings of the IEEE/CVF Conference on Computer Vision and Pattern Recognition. 2024.
>
> [10] Makoviychuk, Viktor, et al. "Isaac gym: High performance gpu-based physics simulation for robot learning" arXiv preprint arXiv:2108.10470 (2021).

---

> > ### Comment · Reviewer_4399 · 2025-08-02
> >
> > Thanks for the detailed and thoughtful response. Most of my concerns have been addressed, and I will raise my rating accordingly. I encourage the authors to incorporate their discussion on robotic articulated object manipulation and humanoid robots into the final version of the paper, as these discussions would further strengthen the soundness of this work.

---

> > > ### Author Response · Authors · 2025-08-03
> > >
> > > Thank you for your encouraging feedback and for raising your rating. We are glad that our responses addressed your concerns. We will carefully incorporate the discussion on robotic articulated object manipulation and humanoid robots into the final version to further improve the clarity and completeness of the work.

---

### Official Review · Reviewer_JkmY · 2025-06-30

**Clarity:** 4
**Significance:** 3
**Originality:** 2
**Rating:** 4
**Confidence:** 3

**Summary:**

The paper aims to address a new task that generates whole-body human-object interaction motions with articulated objects. Leveraging human-object interaction and human-body locomotion data, the paper proposes to split the interaction generation problem into loss function generation and whole-body diffusion noise optimization. Experiments show that the method outperforms several existing methods, and they demonstrate various method applications.

**Questions:**

1) Can the idea of the proposed method apply to the existing human-object interaction generation tasks (e.g., whole-body interaction with rigid objects, hand-object interaction with articulated objects)? If so, the contribution of the paper could increase.

2) In Lines 291-292, the paper points out that the method allows control over keyframe object poses. The object trajectory generation is a simple conditional diffusion model. What is setting optimization targets?

3) What happens if we set the object trajectory or the text as the condition of the hand diffusion models? Is it better than the existing method design?

4) What is the time cost of the proposed method?

**Ethical Concerns:**

["NO or VERY MINOR ethics concerns only"]

**Final Justification:**

The authors' rebuttal has addressed most of my concerns, and I raised my score to borderline accept accordingly.

**Limitations:**

Yes

**Paper Formatting Concerns:**

No.

**Quality:**

2

**Strengths And Weaknesses:**

Strengths:

1) The paper is solving a new and challenging task. Existing interaction generation methods either focus on interacting with rigid objects or neglect human body movements.
2) The proposed method is straightforward.
3) The evaluation metrics comprehensively reflect motion naturalness, motion diversity, and physical plausibility.
4) The ablation study demonstrates the value of individual method designs.
5) The writing is clear and easy to follow.

Weaknesses:

1) The paper claims that the method allows the use of hand-only datasets to generate whole-body motion, whereas the body diffusion model requires motion priors from existing human-body datasets.
2) Head orientation is often unnatural in the qualitative results of the method (e.g., Figure 1(d)). This could be due to the data used to train the body diffusion model is not fully interaction-aware. It can be challenging to generate coordinated and natural bodies and hands using only the gradient descent without the coordination of motion semantics.
3) The time consumption of the method could be significantly large. This limits the real application of the method.
4) Traditional diffusion-based methods (e.g., OMOMO) directly perform one forward diffusion process during inference time, which does not take good physical plausibilities into consideration. The paper's method defines physical loss functions and performs test-time optimization, which ought to achieve better physical performance. Comparing the method with physics-aware baselines (e.g., OMOMO + physics-based post optimization, OMOMO + diffusion noise optimization) is necessary.
5) In the existing baseline methods, only IMoS aims to solve the whole-body interaction generation problem. The paper misses more advanced whole-body interaction generation baselines such as [1-2].
6) Some method names (e.g., "HOIFHLI" in Line 115) are unofficial and could be misleading. It is better to use author names (e.g., "Wu et.al.").

References:
[1] Wu Z, Li J, Xu P, et al. Human-object interaction from human-level instructions[J]. arXiv preprint arXiv:2406.17840, 2024.
[2] Tendulkar P, Surís D, Vondrick C. Flex: Full-body grasping without full-body grasps[C]//Proceedings of the IEEE/CVF Conference on Computer Vision and Pattern Recognition. 2023: 21179-21189.

---

> ### Author Rebuttal · Authors · 2025-07-29
>
> ## **Response to Reviewer JkmY**
> Thank you for your valuable feedback and helpful suggestions. We are glad you found our paper $\textcolor{blue}{\text{clear}}$, $\textcolor{blue}{\text{comprehensive evaluations}}$ and $\textcolor{blue}{\text{well-designed ablations}}$, addressing a $\textcolor{blue}{\text{new and challenging task}}$ with a $\textcolor{blue}{\text{straightforward approach}}$.
>
> ### **Weakness**
> **Using hand-only datasets to generate whole-body motion.**
>
> As noted in $\textcolor{green}{\text{L295}}$ and $\textcolor{green}{\text{L289}}$ of the main paper, our method supports generating whole-body motion from hand-only datasets. Specifically, we refer to two cases:
> 1. $\textcolor{green}{\text{L295}}$ shows that ground-truth hand-only end-effector positions can be directly used to generate whole-body motion through our pipeline.
> 2. $\textcolor{green}{\text{L289}}$ describes that once our model is fully trained, a new hand-only dataset can be used to train only the end-effector trajectory generation model, without re-training the separate diffusion models responsible for whole-body optimization.
>
> Our framework supports the integration of new hand-only data either as direct input or by training an end-effector model. Since the motion models used in the optimization stage remain fixed, the method is highly reusable and extensible. As shown in $\textcolor{green}{\text{Figure 1\(c\)}}$ and $\textcolor{green}{\text{Figure 5}}$ of the main paper, our approach can be easily applied to such hand-only datasets without requiring additional full-body supervision. We will clarify this point in the final version.
>
> **Unnatural head orientation and the coordination between the body and hands.**
>
> This issue occurs only rarely in our results. Moreover, it is easy to address within our framework by incorporating a simple head orientation loss that encourages the head to face the object. We find that adding this term significantly improves head alignment. We thank the reviewer for pointing out this issue.
>
> Regarding the concern on coordination between the body and hands, although the diffusion models are trained separately, they are trained on paired body–hand data from GRAB and ARCTIC and are optimized jointly during inference. The optimization propagates gradients through the kinematic chain, ensuring that fine-grained hand motion objectives influence the global body posture, and vice versa. This design enables natural coordination between the body and the hands.
>
> Furthermore, the end-effector tracking loss plays a central role in enforcing this coherence. Since the hand model controls the high-frequency finger details and the body model influences low-frequency global motion, successfully matching end-effector trajectories requires tight harmonization between all three models. As shown in our experiments ($\textcolor{green}{\text{Tables 1, 3, and 4}}$ of the main paper), this leads to motions with superior physical plausibility, contact quality, and realism, outperforming baselines. We also note that our ablation study ($\textcolor{green}{\text{Table 4, row (e)}}$ of the main paper) shows a performance drop when replacing the decoupled model with a single unified model, confirming that our separate model design, when paired with coordinated optimization, is both effective and beneficial.
>
> While preserving the coordination, the separate model design also enables compositional generation of diverse and flexible whole-body motions, which would be difficult to achieve with a single model.
>
> **Time efficiency.**
>
> We acknowledge that our method is slower than existing approaches, as discussed in $\textcolor{green}{\text{L297}}$ of the main paper. However, our method produces substantially better results than baselines, as demonstrated in $\textcolor{green}{\text{Table 1}}$ and $\textcolor{green}{\text{Table 2}}$ of the main paper. As the first work addressing whole-body manipulation of articulated objects, our primary focus is on achieving high-quality motion. Improving efficiency is an important direction for future work.
>
>
> **Comparison with physics-aware baselines.**
>
> | Methods      | FID ↓ | R-Precision ↑ | Diversity → | Foot skating ↓ | IV ↓  | ID ↓  | CR ↑  |
> |--------------|--------|----------------|--------------|----------------|-------|-------|--------|
> | OMOMO | 3.710 | 0.406 | 6.110 | 0.028 | 13.77 | 15.16 | 0.061 |
> | OMOMO+DNO | 2.873 | 0.391 | 7.004 | 0.022 | 8.95 | 13.30 | 0.075 |
> | Ours     | **2.283** | **0.477**   | **7.208**    | **0.002**      | **5.25** | **12.87** | **0.086** |
>
> Following your suggestion, we apply diffusion noise optimization (DNO) to OMOMO using the same loss functions as in our method. The results are shown in the above table. While noise optimization improves OMOMO, OMOMO+DNO is still worse than our full pipeline. This is because DNO is applied to a whole-body motion model, which we find cannot always produce arm motions consistent with the generated hand trajectory. In contrast, our decoupled design, with separate motion models for body and hands, can leverage more data and achieve better performance. This is also supported by our ablation in $\textcolor{green}{\text{Table 4(e)}}$ of the main paper.
>
> **Comparison with more whole-body interaction baselines [1, 2].**
>
> 1. Wu et al. [1]: The code was not available at the time of submission. Although it was released last week, it remains incomplete, lacking full pipeline training scripts and checkpoints. We will include comparisons once the code is fully released.
> 2. FLEX [2]: This method focuses on whole-body grasp generation rather than interaction motion. Many of its components are specifically designed for achieving stable grasps and are not directly applicable to articulated-object manipulation tasks like ours.
>
> **Reference method naming.**
> The name “HOIFHLI” was borrowed from the original project website of the paper. We will revise it to “Wu et al.” in the final version.
>
> ### **Questions**
> **Applicability to existing human-object interaction generation tasks.**
>
> We have evaluated our method on the GRAB dataset for whole-body interaction with rigid objects, as shown in $\textcolor{green}{\text{Table 3}}$ of the main paper. The visualization results can also be found in the supplementary material. Our method is easily extensible to hand-object interaction settings, which can be seen as a special case of whole-body interaction. We will explore this further in future work.
>
> **Details of controlling keyframe object poses.**
>
> $\textcolor{green}{\text{Section E.2}}$ in the supplementary material presents details related to object motion control. Here, we additionally provide the specific optimization loss used in our setup. We apply diffusion noise optimization using the object motion generation model. Given a set of keyframes where the object pose $\mathbf{S}_j$ is known at certain frames $j$, we define the loss:
>
> $$\mathcal{L}_{\text{keyframe}} = \sum^{j \in \mathcal{K}} \| \hat{\mathbf{S}}_j - \mathbf{S}_j \|^2.$$
>
> Here, $\hat{\mathbf{S}}_j$ denotes the predicted object pose at frame $j$, and $\mathcal{K}$ is the set of keyframe indices. This loss is used to optimize the initial noise of the object motion diffusion model. Once optimized, the generated object trajectory conforms to the specified keyframes, and the resulting end-effector trajectories and whole-body motion follow through our pipeline, producing consistent whole-body behavior for the given keyframe constraints. We will include this explanation in the final version.
>
> **Using object trajectory or text as a condition for the hand diffusion model.**
>
> | Methods      | FID ↓ | R-Precision ↑ | Diversity → | Foot skating ↓ | IV ↓  | ID ↓  | CR ↑  |
> |--------------|--------|----------------|--------------|----------------|-------|-------|--------|
> | w/ condition | 2.998 | 0.453 | **7.212** | 0.002 | 8.95 | 13.30 | 0.075 |
> | Ours     | **2.283** | **0.477**   | 7.208    | **0.002**      | **5.25** | **12.87** | **0.086** |
>
> We experimented with conditioning the hand motion diffusion model on object trajectories. However, as shown in the table, this leads to higher FID. We observe that adding more conditions to the hand model tends to cause overfitting to specific object trajectories, which makes the optimization more difficult. In contrast, our unconditional model provides a stronger and more generalizable motion prior.
>
> **Time cost.**
>
> We report the runtime in $\textcolor{green}{\text{Table 4}}$ of the supplementary material. Our full pipeline takes approximately 17 minutes to generate a 10-second motion clip. While this is slower than the baselines, the improvement in physical realism and motion quality justifies the cost. As the first work tackling whole-body manipulation of articulated objects, our priority is high-quality generation.
>
> ---
> [1] Wu Z, Li J, Xu P, et al. Human-object interaction from human-level instructions[J]. arXiv preprint arXiv:2406.17840, 2024.
>
> [2] Tendulkar P, Surís D, Vondrick C. Flex: Full-body grasping without full-body grasps[C]//Proceedings of the IEEE/CVF Conference on Computer Vision and Pattern Recognition. 2023: 21179-21189.

---

> > ### Author Response · Authors · 2025-08-03
> >
> > Dear Reviewer JkmY,
> >
> > Thank you again for your thoughtful and constructive review. We appreciate your positive comments on the method clarity, evaluation design, and writing quality.
> >
> > We hope our rebuttal has addressed your concerns. Given the short discussion period, we would greatly appreciate it if you could let us know whether our responses have resolved your questions. If there are any remaining concerns or suggestions, we would be grateful for the opportunity to clarify and further improve the paper.

---

> > ### Comment · Reviewer_JkmY · 2025-08-07
> >
> > The rebuttal has addressed most of my concerns and I would raise my score accordingly.

---

> > > ### Author Response · Authors · 2025-08-07
> > >
> > > Thank you for your constructive feedback and for raising your rating. We are glad that our rebuttal addressed your concerns. We will incorporate the clarifications from our rebuttal into the final version.

---

### Official Review · Reviewer_fHcd · 2025-06-30

**Clarity:** 3
**Significance:** 3
**Originality:** 4
**Rating:** 5
**Confidence:** 3

**Summary:**

In this paper, the authors introduce CoDA, a method for generating full-body motion of articulated object manipulation. The method can be used for text-to-motion generation, motion generation using object key frames, and motion generation from hand trajectories. The key insight is to model motion generation of the two hands and body separately, allowing to be supervised by each component's own data source. This is the first method for full-body articulated-object motion generation. Results show superior performance against baselines.

**Questions:**

See "Cons".

**Ethical Concerns:**

["NO or VERY MINOR ethics concerns only"]

**Final Justification:**

After reading the rebuttal and other reviews, I decide to raise the score and champion this paper because the authors have addressed my concerns.

I acknowledge the comments from other reviewers related to limited evaluation (such as limitations in ARCTIC objects). This is the first method that tackles the full-body motion generation with articulated objects. Accepting this paper therefore will encourage future work for more complex interaction setting and thus grows our community for human-object motion generation.

Also, while other reviewers pointed out about robotic applications, this work focuses more on generating animation. So I still think this paper should be accepted.

**Limitations:**

The authors discussed the positive impacts but I think a discussion on the negatives could benefits in a long term. For example, when combined with human pose conditioned video generation methods, this proposed method can be used to generate videos of real individuals (e.g., public figures) performing actions that they did not do. This can raise concerns on misinformation.

**Quality:**

3

**Strengths And Weaknesses:**

## Pros

**The results are impressive**: They are both quantitatively and qualitatively significantly better than the baselines.

**Comprehensive experiments**: The authors provide detailed comparison with many existing baselines (see Table 3) in terms of FID, diversity, and physical metrics like foot skating and interpenetration volume etc on both ARCTIC and GRAB. Detailed ablation is provided in Table 4 to ablate the effect of invididual components such as end-effector BPS and AMASS.

**Good presentation**: The writing is easy to understand. The video is intuitive.

## Cons

**Comparison/clarification on interaction sensors**: Overall, I quite like the paper. The authors introduced "end-effector BPS", a distance-based features to represent the positions of joints in the object coordinate systems. In general, there are similar "sensors" or "interaction encoder" in the motion generation line of work. It would be better if the authors could clarify if the choice of such representations actually make a big difference in the results? For example, in ManipNet, they introduced  "Proximity sensor" and "Signed Distance Sensor", how do they differ in performance compared to "end-effector BPS"?

**Quantitative evaluation on unseen object generalization**: Both ARCTIC and GRAB are a bit limited in terms of object diversity. This means, generalization to unseen object is important for future work. I understand that this is an extremely challenging task given the limited data. However, it is best for the authors to provide some quantitative measures for future work to compare against.

## Justification

Overall, I think the paper is well-written, easy to understand. The experiments are quite comprehensive. The only suggestions are the clarification on existing interaction representations (do they really make much difference?). Further, generalization to unseen objects is an important future direction, and it should be comparable.

---

> ### Author Rebuttal · Authors · 2025-07-29
>
> ## **Response to Reviewer fHcd**
> Thank you for your insightful suggestions and feedback. We are glad that you find our $\textcolor{blue}{\text{results impressive}}$, $\textcolor{blue}{\text{with comprehensive experiments}}$ and a $\textcolor{blue}{\text{good presentation}}$. We also appreciate that you $\textcolor{blue}{\text{quite like the paper}}$.
>
>
> ### **Weakness**
> **Comparison/clarification on interaction representations.**
>
> | Methods      | FID ↓ | R-Precision ↑ | Diversity → | Foot skating ↓ | IV ↓  | ID ↓  | CR ↑  |
> |--------------|--------|----------------|--------------|----------------|-------|-------|--------|
> | w/ SDF | 2.350 | 0.422 | 7.064 | 0.003 | 6.69 | 13.97 | 0.086 |
> | Ours     | **2.283** | **0.477**   | **7.208**    | **0.002**      | **5.25** | **12.87** | **0.086** |
>
> We construct a variant that uses a distance field representation similar to the sensors in ManipNet. Specifically, we represent the end-effector trajectory as a 10×10×10 grid of distances in the object’s local coordinate frame, where each voxel stores its distance to the end-effector. We keep the object geometry encoded using BPS and only replace the end-effector representation. As shown in the above table, our design outperforms this variant.
>
> Both representations offer advantages over directly regressing 3D coordinates, as distance-based encodings better capture the spatial relation between the object and the end-effector. However, our approach further improves performance by aligning the end-effector and object representations in the same BPS space, enabling more consistent and effective interaction learning. The grid-based variant lacks this alignment, leading to slightly lower performance.
>
>
> **Quantitative evaluation on unseen object generalization.**
>
> | Methods      | FID ↓ | R-Precision ↑ | Diversity → | Foot skating ↓ | IV ↓  | ID ↓  | CR ↑  |
> |--------------|--------|----------------|--------------|----------------|-------|-------|--------|
> | OMOMO | 44.009 | 0.547 | 6.234 | 0.028 | 26.92 | **8.23** | 0.116 |
> | Ours     | **16.091** | **0.547**   | **4.964**    | **0.002**      | **26.24** | 8.56 | **0.128** |
>
> To evaluate the generalization capability of our method, we conduct an additional experiment on the ARCTIC dataset by holding out the box object for testing and training on the remaining objects. As shown in the above table, our method achieves much better FID compared to the baseline.
>
> ### **Social impact**
> **Concerns about misinformation.**
>
> We agree with the reviewer that generated motion could potentially be used as input to generate fake videos of individuals. We will discuss this concern in the final version.

---

> > ### Author Response · Authors · 2025-08-03
> >
> > Dear Reviewer fHcd,
> >
> > Thank you again for your thoughtful and constructive review. We appreciate your positive comments on the method design, results, and experiments.
> >
> > We hope our rebuttal has addressed your concerns. Given the short discussion period, we would greatly appreciate it if you could let us know whether our responses have resolved your questions. If there are any remaining concerns or suggestions, we would be grateful for the opportunity to clarify and further improve the paper.

---

> > ### Comment · Reviewer_fHcd · 2025-08-03
> >
> > Thank you for your detailed rebuttal. After reading the rebuttal and other reviews, I decide to raise the score and champion this paper.
> >
> > I acknowledge the comments from other reviewers related to limited evaluation (such as limitations in ARCTIC objects). This is the first method that tackles the full-body motion generation with articulated objects. Accepting this paper therefore will encourage future work for more complex interaction setting and thus grows our community for human-object motion generation.

---

> > > ### Author Response · Authors · 2025-08-03
> > >
> > > Thank you for your thoughtful feedback, for raising the score, and for championing our paper. We also sincerely appreciate your kind remarks on the potential impact of accepting this work. We will carefully incorporate the content of our rebuttal into the final version.

---

### Official Review · Reviewer_3mBR · 2025-07-03

**Clarity:** 4
**Significance:** 3
**Originality:** 3
**Rating:** 4
**Confidence:** 4

**Summary:**

This paper proposed a novel coordinated diffusion noise optimization framework for human-objection interactions generation. The specialized hands, body, object diffusion models are trained on its own datasets separately. The proposed unified representation captures fine-grained spatial relationships between the hand and articulated object parts, obtaining highly accurate interaction motion. Extensive experiments show the superiority of the proposed method in quality, physical plausibility, and capabilities.

**Questions:**

1. Will the separate models limit the potential to learn inter-correlation between parts, such as how the object types/movements affect the body motion. For example, if a person will have a less probability to run if he is operating a larger items like microwave, compared with some small items like phone.
2. Is it pipeline limieted to hand manipulation? Is that possible to extend it to whole-body contact?

**Ethical Concerns:**

["NO or VERY MINOR ethics concerns only"]

**Final Justification:**

The authors' rebuttal has addressed most of my concerns, and I have raised my score to accept accordingly.

**Limitations:**

yes

**Quality:**

3

**Strengths And Weaknesses:**

+ The proposed coordinated diffusion noise optimization framework is interesting. It allows to train specialized diffusion models on its own datasets and can merge the expertises together to generate whole-body interactions.
+ The proposed unified BPS-based representation for object and end-effectors is novel and seems effective. It facilitates geometric alignment between end-effectors and object parts.

- Computational Cost and Efficiency. The multi-stage pipeline, especially the optimization process with diffusion, is significantly more computationally intensive than standard generative approaches.

---

> ### Author Rebuttal · Authors · 2025-07-29
>
> ## **Response to Reviewer 3mBR**
> Thank you for your thoughtful feedback and valuable suggestions. We are glad that you found our framework $\textcolor{blue}{\text{interesting}}$ and the proposed unified BPS-based representation $\textcolor{blue}{\text{novel and effective}}$.
>
> ### **Weakness**
> **Computational efficiency.**
>
> We acknowledge that our method is more computationally intensive than existing approaches, as discussed in $\textcolor{green}{\text{L297}}$ of the main paper. However, as the first work targeting whole-body manipulation of articulated objects, our primary focus is more on generating high-quality and physically plausible motion. Our method significantly outperforms prior baselines in terms of motion quality, as shown in $\textcolor{green}{\text{Table 1}}$ and $\textcolor{green}{\text{Table 2}}$ of the main paper. We believe that improving computational efficiency is definitely an important and promising direction for future work.
>
> ### **Questions**
> **Will the separate (body, right/left hand) models limit the potential to learn inter-correlation between parts?**
>
> While our model adopts a decoupled architecture for the body, left hand, and right hand, it does not limit the capacity to learn inter-part coordination. On the contrary, coordination naturally emerges as a result of the paired training data and the inference through our joint optimization process in noise space.
>
> Specifically, although the diffusion models are trained separately, they are trained on paired body–hand data from GRAB and ARCTIC and are optimized jointly during inference. The optimization propagates gradients through the kinematic chain, ensuring that fine-grained hand motion objectives influence the global body posture, and vice versa. This design enables natural coordination between the body and the hands.
>
> Furthermore, the end-effector tracking loss plays a central role in enforcing this coherence. Since the hand model controls the high-frequency finger details and the body model influences low-frequency global motion, successfully matching end-effector trajectories requires tight harmonization between all three models. As shown in our experiments ($\textcolor{green}{\text{Tables 1, 3, and 4}}$ of the main paper), this leads to motions with superior physical plausibility, contact quality, and realism, outperforming baselines. We also note that our ablation study ($\textcolor{green}{\text{Table 4, row (e)}}$ of the main paper) shows a performance drop when replacing the decoupled model with a single unified model, confirming that our separate model design, when paired with coordinated optimization, is both effective and beneficial.
>
> Similarly, the inter-correlation between the object and the body is implicitly learned through the paired data. Manipulating a large object like a microwave often results in simple and stationary object motion and end-effector trajectories, leading to relatively static body motion. In contrast, smaller objects like a phone typically involve more dynamic object motion and end-effector trajectories, which in turn produce more active whole-body responses.
>
> The advantage of using separate models rather than a single model as the motion prior is that it allows us to leverage more diverse and part-specific datasets. As shown in $\textcolor{green}{\text{Figure 1\(c,d\)}}$ of the main paper, this design supports a richer combination of motion patterns and enables compositional generation of whole-body behaviors.
>
> **Is the pipeline limited to hand manipulation? Is it possible to extend it to whole-body contact?**
>
> Our pipeline is not limited to hand manipulation. While we currently use the fingertips as the end-effectors, this choice is due to the nature of the ARCTIC dataset, which primarily contains standing manipulation tasks. In principle, the end-effector trajectory model can be extended to include other interacting joints such as elbows, shoulders, knees, or feet. Our optimization framework is highly flexible and agnostic to the number and type of joints selected as control targets. Any set of joints can be designated as optimization targets, and gradients from their tracking losses will propagate through the kinematic chain to guide the whole-body motion generation in a coordinated and physically plausible manner.

---

> > ### Author Response · Authors · 2025-08-03
> >
> > Dear Reviewer 3mBR,
> >
> > Thank you again for your thoughtful and constructive review. We appreciate your positive comments on our framework design and unified BPS-based representation.
> >
> > We hope our rebuttal has addressed your concerns. Given the short discussion period, we would greatly appreciate it if you could let us know whether our responses have resolved your questions. If there are any remaining concerns or suggestions, we would be grateful for the opportunity to clarify and further improve the paper.

---

> > > ### Comment · Reviewer_3mBR · 2025-08-08
> > > **Response to authors**
> > >
> > > Thanks to the authors for the clarification. It clearly resolves my concerns.

---

> > > > ### Author Response · Authors · 2025-08-09
> > > >
> > > > Thank you for confirming that our rebuttal addressed your concerns and for your constructive feedback. We sincerely appreciate your support and recognition of our work, and we will carefully incorporate the clarifications from the rebuttal into the final version.

---

### Decision · Program_Chairs · 2025-09-17

**Decision:**

Accept (poster)

**Comment:**

This paper proposes CoDA, the first method for synthesizing whole-body manipulation of articulated objects by coordinating body, hand, and object motions through joint noise-space optimization of three separate diffusion models.

The authors' rebuttal effectively addressed reviewers' concerns, leading to improved scores. Both reviewers and the AC recognize the significance of this novel task in advancing human-object interaction research, and expect this paper to encourage future work in more complex interaction scenarios, thereby expanding our community's capabilities in motion generation.

While reviewers noted concerns about computational cost and efficiency, we believe these limitations are outweighed by the method's innovative contributions and potential for future optimization. The paper's strong technical merits and the importance of the addressed problem justify acceptance.